# Strong negative nonlinear friction from induced two-phonon processes in vibrational systems

X. Dong [1,2], M.I. Dykman [3] & H.B. Chan[1,2]

Self-sustained vibrations in systems ranging from lasers to clocks to biological systems are often associated with the coefficient of linear friction, which relates the friction force to the velocity, becoming negative. The runaway of the vibration amplitude is prevented by positive nonlinear friction that increases rapidly with the amplitude. Here we use a modulated electromechanical resonator to show that nonlinear friction can be made negative and sufficiently strong to overcome positive linear friction at large vibration amplitudes. The experiment involves applying a drive that simultaneously excites two phonons of the studied mode and a phonon of a faster decaying high-frequency mode. We study generic features of the oscillator dynamics with negative nonlinear friction. Remarkably, self-sustained vibrations of the oscillator require activation in this case. When, in addition, a resonant force is applied, a branch of large-amplitude forced vibrations can emerge, isolated from the branch of the ordinary small-amplitude response.

---

[1] Department of Physics, The Hong Kong University of Science and Technology, Clear Water Bay, Kowloon, Hong Kong, China. [2] William Mong Institute of Nano Science and Technology, The Hong Kong University of Science and Technology, Clear Water Bay, Kowloon, Hong Kong, China. [3] Department of Physics and Astronomy, Michigan State University, East Lansing, MI 48824, USA. Correspondence and requests for materials should be addressed to H.B.C. (email: hochan@ust.hk)

Nonlinear friction[1,2] was first used by van der Pol to describe, phenomenologically, the operation of radio-frequency generators[3]. In the van der Pol model, the energy gain is described by negative linear friction. The nonlinear friction is positive, corresponding to the rate of energy loss that increases superlinearly with the energy of the vibrations. This model has been broadly used in science and engineering. Recently, there has been renewed interest in positive nonlinear friction following its observations in various passive (rather than self-oscillating) nano-, micro-, and optomechanical systems[4–10]. Furthermore, such friction has also been engineered in microwave cavities in order to create long-lived coherent quantum states[11,12].

A simple microscopic mechanism of linear friction is a decay process where a single energy quantum of the vibrational mode (a phonon) goes into excitations of a thermal reservoir[13]. In contrast, positive nonlinear friction originates from the decay that involves two vibrational quanta of the mode[14]. For nonlinear friction of nanomechanical modes and cavity modes, the excitations of the reservoir can be, respectively, phonons[15,16] or propagating photons[11] (Fig. 1a). For the cavity mode studied by Leghtas et al.[11], the two-photon decay was stimulated by a pump. This process resembles the stimulated decay in optomechanics[17] where an input photon leads to the removal or addition of only one quantum of the mechanical mode. Provided that the cavity

**Fig. 1** Two-phonon transfer between the plate and beam modes of the electromechanical resonator. **a** Ordinary positive nonlinear friction involves transferring two phonons of the vibrational mode 1 (shown by the wavy lines) to excitations of the reservoir (the dashed line). **b** External pump at frequency $\omega_F$ (the solid line) can create one phonon in fast-decaying mode 2 and two phonons in mode 1, inducing negative nonlinear friction in mode 1. **c** Colorized scanning electron micrograph of the electromechanical resonator with a schematic of the measurement circuitry. The white scale bar at the lower right corner measures 20 μm. **d** The amplitude $a_1$ of the plate mode (mode 1) against frequency $\omega_{d1}$ of a small probe voltage (probe 1). The dots are measurements. The line is a fit to yield parameters $\Gamma_1$ and $\Gamma_2$. Inset: vibration profile of the plate mode. The color bar gives the normalized displacement. **e** A similar plot for the beam mode (mode 2). **f** A sketch of the spectra of the plate mode and the beam mode centered at the mode frequencies $\omega_1$ and $\omega_2$, respectively. Pumping at the secondary high-/low-frequency sidebands (red and blue arrows) generates negative/positive nonlinear friction in the plate mode

mode decays much faster than the mechanical mode, such process is induced by pumping at a sideband of the cavity eigenmode[18–20]. If the pump frequency is red-detuned by the frequency of the mechanical mode, the linear friction coefficient of the mechanical mode increases and the mode temperature decreases. This backaction mechanism enables observation of interesting quantum effects with nanomechanical systems[17,21,22]. On the other hand, blue-detuned pumping, if sufficiently strong, makes the linear friction coefficient negative, exciting self-sustained oscillations[23–30]. However, to our knowledge, friction that becomes negative only for sufficiently large vibration amplitudes, i.e., absolute negative nonlinear friction, has not been obtained in optomechanics.

We note that the decrease of losses with increasing vibration energy was reported for both microwave cavities[31,32] and nano-mechanical systems[33]. It was attributed to absorption saturation in two-level systems that absorb the energy from the vibrations. By its nature, such nonlinear friction cannot make the overall energy loss negative. However, it plays an important role in providing a means to increase the quality factor of super-conducting microwave cavities used in optomechanics[34,35].

In this work we show that a properly designed pumping generates strong negative nonlinear friction, and we study the qualitatively new features of vibrational dynamics that come with such friction. The experiment is done on a micro-mechanical resonator with two nonlinearly coupled vibrational modes of strongly differing frequencies $\omega_{1,2}$ and linear decay rates $\Gamma_{1,2}$, with $\omega_2 \gg \omega_1$ and $\Gamma_2 \gg \Gamma_1$. Pumping at the blue-detuned secondary sideband of the higher-frequency mode $\omega_2 + 2\omega_1$ opens a relaxation channel where two quanta of mode 1 and one quantum of mode 2 are simultaneously excited (Fig. 1b). The resulting negative nonlinear friction of mode 1 can be controlled and made strong enough to overcome the intrinsic positive linear friction. We find that the modes can then settle into a state of stable self-sustained vibrations. However, these vibrations have to be activated, i.e., require a sufficiently strong initial excitation. At weaker pump power where the negative nonlinear friction is not strong enough to make the full friction force negative even for larger amplitudes, it still significantly modifies the oscillator dynamics in the presence of resonant drive. On top of the ordinary Lorentzian-type dependence of the amplitude of forced vibrations on the drive frequency, there emerges an additional branch of large-amplitude vibrations. Unlike the familiar nonlinear response of vibrations with conservative nonlinearity, this branch is disconnected from the small-amplitude branch. It emerges already for a comparatively weak drive. When the frequency of the drive or the mode varies, there occur jumps to the small-amplitude branch at both ends of the high-amplitude branch, opening opportunities in detecting perturbations to the system of both polarity, in contrast to bifurcation amplifiers based on the conservative nonlinearity[36–38].

## Results

### The electromechanical resonator

Our resonator consists of three parts (Fig. 1c). The first is a polycrystalline silicon plate with dimension $100 \times 100 \times 3.5\,\mu m$. It is supported on its opposite sides by two silicon beams (1.3 μm wide and 2 μm thick). The two beams have different lengths of 80 and 75 μm. Both of them are coated with 30 nm of gold. The plate performs vibrations normal to the substrate at eigenfrequency $\omega_1 = 272599.72\,rad\,s^{-1}$. As seen from Fig. 1d, the damping constant of this vibrational mode, which we call mode 1, is $\Gamma_1 = 3.26\,rad\,s^{-1}$. The system also has a mode in which the longer beam vibrates parallel to the substrate (Fig. 1e). We call it mode 2. It has a much higher frequency $\omega_2 =$

9942136.19 rad s$^{-1}$ and a higher damping constant $\Gamma_2 = 187.57$ rad s$^{-1}$.

Modes 1 and 2 are parametrically coupled. This coupling originates from the tension generated in the beam as the plate moves normal to the substrate, which in turn modifies the spring constant for the motion of the beam parallel to the substrate, and vice versa. As the plate vibrates, sidebands at frequencies $\omega_2 \pm n\omega_1$ with $n = 1, 2,\ldots$ are created in the spectrum of the response of mode 2 (the beam mode) around its frequency $\omega_2$.

**The model.** To induce nonlinear friction, we pump the system at the secondary sidebands, i.e., the combination frequencies $\omega_2 \pm 2\omega_1$, by applying an ac current to the beam in a magnetic field. Our system is in the deep resolved-sideband limit, with $\omega_1/\Gamma_2 = 1453$. The minimalistic model that captures the resonant behavior is described by equations

$$\ddot{q}_1 + \omega_1^2 q_1 + 2\Gamma_1 \dot{q}_1 + (\gamma/m_1)q_1 q_2^2 + (\gamma_1/m_1)q_1^3$$
$$= \left(F_p/m_1\right)2q_1 q_2 \cos(\omega_F t) \tag{1}$$

$$\ddot{q}_2 + \omega_2^2 q_2 + 2\Gamma_2 \dot{q}_2 + (\gamma/m_2)q_1^2 q_2 + (\gamma_2/m_2)q_2^3$$
$$= \left(F_p/m_2\right)q_1^2 \cos(\omega_F t) \tag{2}$$

where $q_{1,2}$ are, respectively, the displacement of the plate and the midpoint of the beam, $m_{1,2}$ are the effective masses of the two modes, $\gamma$ denotes the dispersive coupling coefficient (the coupling energy is $\frac{1}{2}\gamma q_1^2 q_2^2$), and $\gamma_{1,2}$ are the coefficients of the Duffing nonlinearity of the two modes. Parameter $F_p$ determines the effective amplitude of near-resonant parametric pumping at frequency $\omega_F$ close to $\omega_2 \pm 2\omega_1$. It corresponds to the term $-F_p q_1^2 q_2 \cos(\omega_F t)$ in the modes Hamiltonian. This term effectively incorporates, in a standard way, the contribution from the linear force ($\propto \cos\omega_F t$) weighted with the parameters of the nonlinear non-resonant mode coupling.

For pumping frequencies close to the upper combination frequency, $|\omega_F - 2\omega_1 - \omega_2| \ll \omega_{1,2}$, we change from $q_{1,2}(t), \dot{q}_{1,2}(t)$ to dimensionless complex amplitudes $v_{1,2}(t)$ defined as $v_1(t) = (m_1/2\omega_1 C_{sc})^{1/2}[\omega_1 q_1(t) - i\dot{q}_1(t)]\exp(-i\omega_1 t)$, $v_2(t) = (m_2/2\omega_2 C_{sc})^{1/2} [(\omega_F - 2\omega_1)q_2(t) - i\dot{q}_2(t)]\exp[-i(\omega_F - 2\omega_1)t]$. The scaling constant $C_{sc}$ has the dimension of action and can be chosen as the energy of mode 1 divided by its frequency; we set $C_{sc} = 10^{-21}\,J \cdot s$, which corresponds to a characteristic displacement ~10 nm. The vibration amplitudes $A_{1,2}$ are expressed in terms of the variables $v_{1,2}$ as $A_i = (2C_{sc}/m_i\omega_i)^{1/2}|v_i|$ with $i = 1, 2$. In the rotating wave approximation, equations for $v_{1,2}(t)$ read

$$\dot{v}_1 + \Gamma_1 v_1 = i\partial H_{RWA}/\partial v_1^* \tag{3}$$

$$\dot{v}_2 + \Gamma_2 v_2 = i\partial H_{RWA}/\partial v_2^* \tag{4}$$

$$H_{RWA} = -\Delta|v_2|^2 + \Lambda_{12}|v_1|^2|v_2|^2 + \frac{1}{2}\sum_{i=1,2}\Lambda_{ii}|v_i|^4 - f_p(v_1^2 v_2 + c.c.) \tag{5}$$

Here $H_{RWA}$ is the Hamiltonian of the driven modes in the rotating wave approximation. Parameter $\Delta = \omega_F - 2\omega_1 - \omega_2$ is the frequency detuning from the combination resonance, $|\Delta| \ll \omega_1$. Parameters $f_p$ (18.332 s$^{-1}$), $\Lambda_{11}$ (2.201 s$^{-1}$), $\Lambda_{22}$ (1627.7 s$^{-1}$), and $\Lambda_{12}$ (33.234 s$^{-1}$) are related to $F_p$, $\gamma$, $\gamma_1$, and $\gamma_2$ in Eq. (1), respectively (Supplementary Note 2). However, in general they also have contributions from the cubic nonlinearity

of the resonator modes disregarded in Eqs. (1) and (2). Importantly, $\Lambda_{ij}$ are the parameters that can be directly measured, for example, by applying periodic drives to the modes (Supplementary Note 1), whereas $f_p$ is externally controlled.

In the case of driving close to the lower combination frequency, $|\omega_F - \omega_2 + 2\omega_1| \ll \omega_{1,2}$, one sets $v_2(t) = (m_2/2\omega_2 C_{sc})^{1/2} [(\omega_F + 2\omega_1)q_2(t) - i\dot{q}_2(t)]\exp[-i(\omega_F + 2\omega_1)t]$. Equations for $v_{1,2}$ have the same form as Eqs. (3)–(5), except that the last term in $H_{RWA}$ now reads $-f_p(v_1^2 v_2^* + \text{c.c.})$ and $\Delta = \omega_F + 2\omega_1 - \omega_2$.

**Weak to moderate negative nonlinear friction**. We first explore the energy dissipation for not too large amplitudes of mode 1 (the plate mode), where the nonlinear friction is moderately strong. A familiar description of nonlinear friction[4–8,31,33] is obtained for $\Gamma_2 \gg \Gamma_1$ where mode 2 serves as a thermal reservoir for mode 1. The analysis simplifies if the dispersive mode coupling and the internal nonlinearity of the modes are small, so that the nonlinearity-induced frequency shifts are small compared to $(\Gamma_2^2 + \Delta^2)^{1/2}$. Then, after a short transient we have $|\dot{v}_1/v_1|, |\dot{v}_2/v_2| \ll (\Gamma_2^2 + \Delta^2)^{1/2}$, and we can apply a simple adiabatic approximation by setting $\dot{v}_2 = 0$ in Eq. (2) (a complete analysis is given in Supplementary Note 2). In this approximation Eqs. (3)–(5) yield:

$$\dot{v}_1 \approx -v_1(\Gamma_1 + \alpha|v|^2) + i\beta v_1|v_1|^2 \quad (6)$$

where $\alpha = -2f_p^2\Gamma_2/(\Gamma_2^2 + \Delta^2)$ and $\beta = \Lambda_{11} + 2f_p^2\Delta/(\Gamma_2^2 + \Delta^2)$ ($\alpha = -1.509\ \text{s}^{-1}$ and $\beta = 0.4326\ \text{s}^{-1}$ for $\Delta = -35\ \text{Hz}$). In this case $\Gamma_{ad} = \Gamma_1 + \alpha|v_1|^2$ represents a decay rate for mode 1. For the negative nonlinear friction considered ($\alpha < 0$), this rate is decreased from $\Gamma_1$ by a parameter quadratic in the vibration amplitude. For driving at the lower secondary sideband, the sign of $\alpha$ is reversed, leading to the conventional positive nonlinear friction.

To measure the decay rate, we employ the standard ringdown technique. We first drive mode 1 into a steady state of vibrations with a resonant periodic force. Then, we turn off the periodic force and record the decrease in vibration amplitude $A_1 = (m_1\omega_1/2C_{sc})^{-1/2}|v_1|$ as a function of time, as shown in Fig. 2a. When there is no sideband pumping, $A_1$ decays exponentially as expected (black dots). The slope of the semi-logarithmic plot gives the nearly instantaneous (on the timescale $\gg \omega_1^{-1}$) decay rate $\Gamma_{inst} = \text{dlog}A_1/\text{d}t$ that is independent of the vibration amplitude, as shown in Fig. 2b. Weak nonlinear friction is induced by turning on the sideband pumping and choosing the detuning frequency $\Delta$ to be relatively large ($\sim-35$ Hz). As shown by the blue/red curves in Fig. 2a, b for pumping at the upper/lower sidebands, the decay rate decreases/increases with vibration amplitude giving negative/positive nonlinear friction. At small vibration amplitudes, measurements agree well with the dashed lines that represent Eq. (6), $\Gamma_{inst} \approx \Gamma_1(1 + \tilde{\alpha}A_1^2)$ with $\tilde{\alpha} = \alpha m_1\omega_1/2C_{sc}$. Deviations become apparent at larger amplitudes, where the conservative nonlinearity of the modes and their dispersive coupling need to be taken into account (Supplementary Note 2).

The decay significantly changes for larger initial vibration amplitudes. Here the dependence of the instantaneous decay rate on vibration amplitude becomes non-monotonic. Figure 2c, d shows respectively the measured decay in time and the corresponding instantaneous decay rate as a function of the vibration amplitude. For pumping at the blue secondary sideband, $\Gamma_{inst}$ attains a minimum and then approaches the zero-amplitude value $\Gamma_1$ as the amplitude further increases. This behavior is a consequence of the conservative nonlinearities of the coupled modes disregarded in Eq. (6). At large amplitudes, the amplitude-dependent frequencies $\omega_{1,2}(|v_1|^2, |v_2|^2)$ no longer satisfy the resonant condition for nonlinear friction $\omega_F \approx \omega_2 \pm 2\omega_1$. Therefore, the decay rate approaches the linear

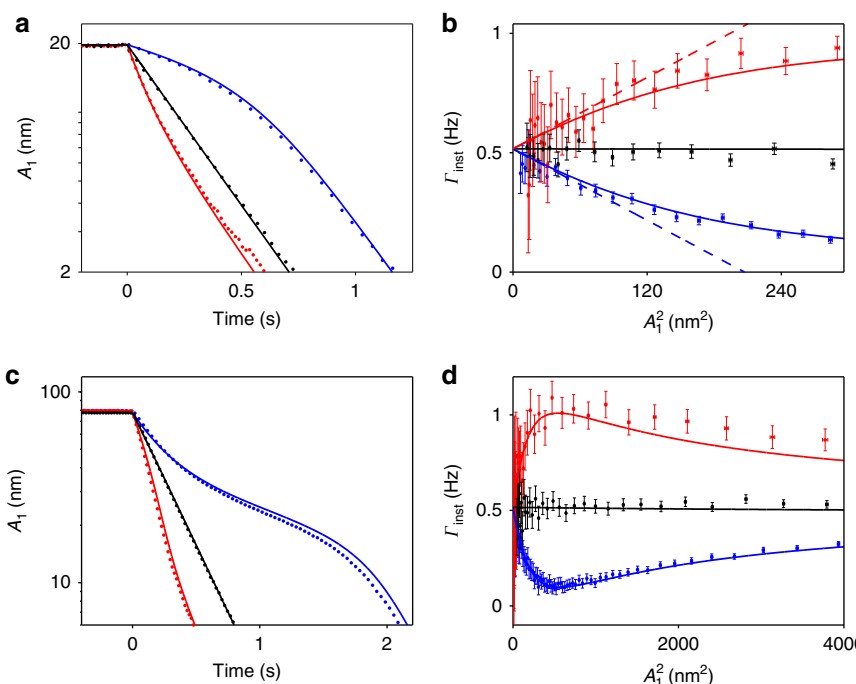

**Fig. 2** Positive and negative nonlinear friction on the plate mode due to two-phonon processes. **a** Measured (dots) and calculated (lines) ringdown of the dimensional vibration amplitude $A_1$ of mode 1 with no sideband pumping (black) and pumping at the red-/blue-detuned secondary sideband (red/blue) that leads to positive/negative nonlinear friction; $\Delta = -35$ Hz. **b** The instantaneous decay rate as a function of the squared vibration amplitude. Dashed lines are from Eq. (6), solid lines show the extended adiabatic theory (Supplementary Note 2). **c**, **d** Similar plots as (**a**) and (**b**), for larger initial vibration amplitude. Error bars represent ±1 s.e.

decay rate $\Gamma_1$, as shown in Fig. 2d. Calculations are plotted as solid lines and are in good agreement with measurement (Supplementary Note 2).

**Self-sustained vibrations**. When the pump detuning $|\Delta|$ at the blue-detuned secondary sideband is reduced, negative nonlinear friction becomes stronger. The minimum in the instantaneous decay rate as a function of $|\nu_1|$ becomes deeper and eventually drops below zero in a certain range of amplitudes. This leads to a novel type of self-sustained vibrations. They are qualitatively different from the self-sustained vibrations in opto- and nano-mechanics that emerge when the pumping makes the coefficient of linear friction negative[18,23–30]. A major difference is that the excitation of self-sustained vibrations by negative nonlinear friction requires activation. Nonlinear friction has no effect when the vibration amplitude is small. If the initial vibration amplitude is zero, self-sustained vibrations cannot be excited by merely sweeping the pump power or the pump frequency. This behavior is seen from the lower branch of data in Fig. 3a. If, however, the vibration amplitude is perturbed beyond certain threshold, the system can settle into a state where it performs self-sustained vibrations. Their typical spectrum is shown in the inset of Fig. 3a. A change in the pump detuning $\Delta$ leads to a change in the vibration amplitude, as shown by the upper branch of data in Fig. 3a. As $\Delta$ is decreased beyond the bifurcation point $\Delta_B$ ($\approx -24.1$ Hz), the vibration amplitude jumps discontinuously to zero. The system cannot return to the self-sustained vibration state unless the amplitude is perturbed sufficiently strongly from zero by a source different from the pump. This type of bistability qualitatively differs from the well-known bistability of resonantly or parametrically modulated Duffing oscillators[37–39] or coupled modes pumped close to the sum frequency[23–30].

Figure 3b shows that self-sustained vibrations of the plate mode (mode 1) are accompanied by vibrations of the beam mode (mode 2). In the presence of fluctuations, the vibrations of both modes undergo phase diffusion, with remarkable properties associated with the discrete time-translation symmetry imposed by the pump[30,40] that are, however, outside the scope of this paper.

Figure 3c shows how the plate mode settles into the zero-amplitude state or the self-sustained vibration state (with amplitude $A_{st}$) depending on the initial vibration amplitude, with $\Delta$ fixed at 0 Hz. For the initial amplitude smaller than the threshold value $A_{th}$ (purple circles), the negative nonlinear friction is smaller than the positive linear friction. Vibrations ring down toward zero in a non-exponential manner. For the initial amplitude lying between $A_{th}$ and $A_{st}$, the negative nonlinear friction overcomes the positive linear friction. By absorbing energy from the pump through two-phonon processes, the plate mode rings up in amplitude toward $A_{st}$. For the initial amplitude larger than $A_{st}$ (dark red circles), the overall friction is positive. The vibration amplitude decays, but toward $A_{st}$ instead of zero. Thus, $A_{th}$ represents the threshold amplitude for self-sustained vibrations to develop. Different pump detuning yields different values of $A_{th}$. The measured value of this threshold is plotted in light red in Fig. 3a. As the pump detuning frequency $\Delta$ approaches the bifurcation value $\Delta_B$, the amplitudes $A_{st}$ and $A_{th}$ merge, leading to a discontinuous jump of the vibration amplitude to zero.

Near the bifurcation point, the dynamics can be mapped onto a saddle-node bifurcation of the radius of the limit cycle (Supplementary Note 3). We note that the actual separation of the regions of attraction to the coexisting states of self-sustained vibrations and the zero-amplitude state occurs on a hypersurface in the four-dimensional space of the dynamical variables of modes 1 and 2. The value of $A_{th}$ is the projection of this hypersurface on the plane of variables of mode 1. In the

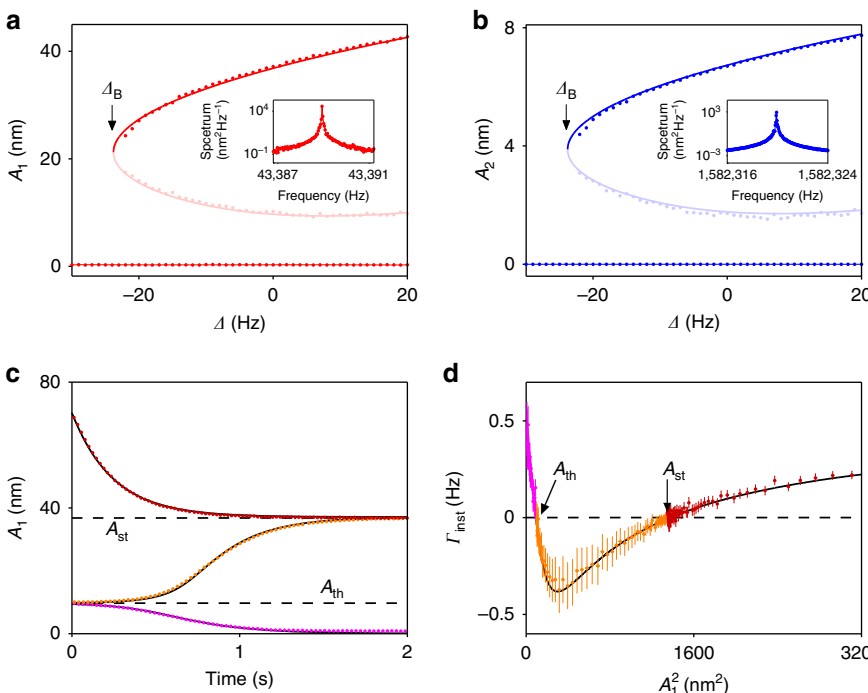

**Fig. 3** Self-sustained vibrations associated with strong negative nonlinear friction. **a** Amplitude of vibrations of mode 1 as a function of pump detuning $\Delta$. The stable branches are in red; they correspond to self-sustained vibrations and a zero-amplitude state. The light red dots represent the threshold amplitude for exciting self-sustained vibrations. The error bar is smaller than the dot size. Inset: spectrum of the self-sustained vibrations. **b** The same plot for mode 2. **c** Depending on the initial amplitude, mode 1 decays to zero (purple) or settle into self-sustained vibrations (orange and dark red). The two dashed lines mark the values of $A_{st}$ and $A_{th}$. **d** The instantaneous decay rate extracted from (**c**). The solid lines in all panels are theoretical predictions (Supplementary Notes 2 and 3). Error bars represent ±1 s.e.

experiment, it is obtained by driving mode 2 and following the dynamics of mode 1 after the drive is turned off; $A_{th}$ is independent of the phase of mode 1, since $\omega_1$ is incommensurate with the pump frequency.

Figure 3d illustrates the activated nature of self-sustained vibrations in an alternative way, by plotting the instantaneous decay rate $\Gamma_{inst}$ as a function of the vibration amplitude $A_1$. This plot bears resemblance to the blue curve in Fig. 2d recorded at larger $|\Delta|$, except that the minimum has dropped below zero due to strong negative nonlinear friction. The instantaneous decay rate in Fig. 3d crosses zero at two vibration amplitudes $A_{th}$ and $A_{st}$ that represent the threshold amplitude and the amplitude for stable self-sustained vibrations, respectively. The theory is in excellent agreement with the measurements.

**Forced vibrations**. An important consequence of nonlinear friction is the change of the spectral response of the mode to a resonant periodic force $F_{d1}\cos\omega_{d1}t$ with $\omega_{d1} \approx \omega_1$[14] and of the dependence of the response on the force amplitude $F_{d1}$ (Methods). In the linear regime, the vibration amplitude attains maximum value $a_{1max}$ when $\omega_{d1} = \omega_1$. With no nonlinear friction the ratio $\Gamma_{peak} = F_{d1}/2m_1\omega_1 a_{1max}$ is constant even in the presence of Duffing nonlinearity[41] (black line in Fig. 4b). With negative nonlinear friction, but with no self-sustained oscillations initiated, the dependence of the vibration amplitude $a_1$ on $\omega_{d1}$ remains Lorentzian for small $F_{d1}$, see light blue curve in Fig. 4a, and $a_{1max}$ is nearly proportional to $F_{d1}$. However, as $F_{d1}$ increases, this measured dependence (dark blue circles in Fig. 4a) becomes sharper than the Lorentzian (dark blue curve in Fig. 4a). The blue data in Fig. 4b show that $\Gamma_{peak}$ decreases with $F_{d1}$. In other

experiments[4–8,31,33], a nonlinear dependence of $a_{1max}$ on $F_{d1}$ was interpreted as the evidence of the presence of positive[4–9] and negative[33] nonlinear friction.

Next we detune the pump frequency so that $\Delta < \Delta_B$ and the overall friction remains positive albeit strongly amplitude-dependent. We find that, in this regime, negative nonlinear friction leads to a qualitatively new branch of the stable states of forced vibrations at the drive frequency $\omega_{d1}$. These states are not straightforward to detect. Figure 4c plots the vibration amplitude of the plate (mode 1) as a function of $\omega_{d1}$ at $\Delta = -35$ Hz. The bottom part of the response resembles that of a driven harmonic oscillator, well described by the square root of a Lorentzian. Here the vibration amplitude is small so that negative nonlinear friction has negligible effect on the lineshape. Interestingly, we observe another branch, with amplitude between ~15 and 29 nm in Fig. 4c, where negative nonlinear friction reduces the overall damping and leads to stable oscillations at the drive frequency $\omega_{d1}$, but with much higher amplitude. The driving amplitude for this extra branch is identical to that of the resonance peak of the lower branch. In contrast to the nonlinear response of a Duffing oscillator with linear friction[41], this branch is isolated from the resonance peak at low amplitudes. It exists in the frequency range $\omega_L \leq \omega_{d1} \leq \omega_H$. If the driving frequency is swept beyond the bifurcation values $\omega_{L,H}$, the system jumps back to the low-amplitude branch.

The high-amplitude branch exists because the overall damping is decreased considerably by strong negative nonlinear friction. It cannot be accessed by sweeping the driving frequency alone. A sufficiently large perturbation of the amplitude is required. We used strong driving force to excite vibrations of large amplitude at driving frequencies between $\omega_L$ and $\omega_H$. Upon decreasing the

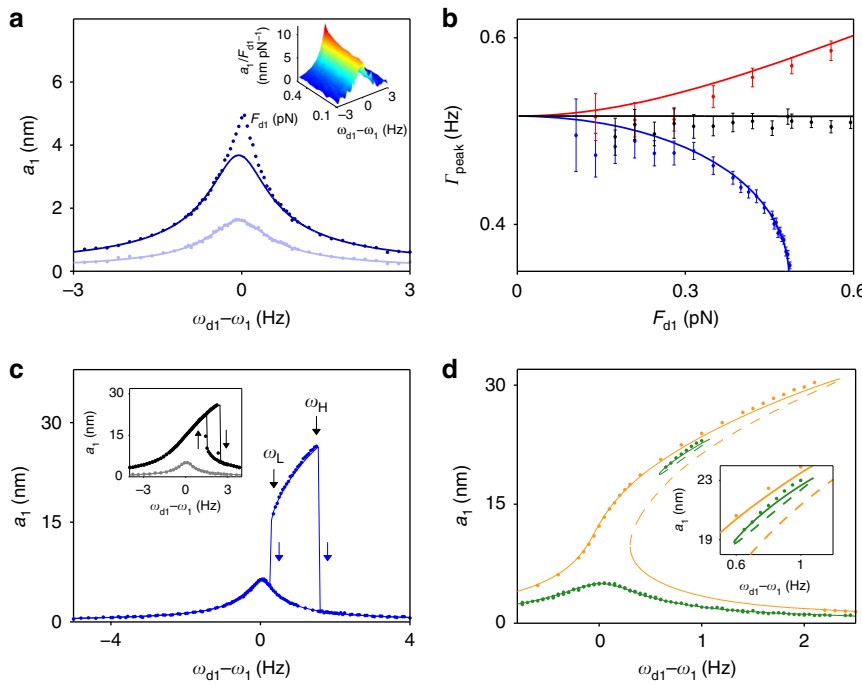

**Fig. 4** Effect of negative nonlinear friction on the resonant response of the plate mode. **a** The amplitude of forced vibrations $a_1$ versus the driving force frequency $\omega_{d1}$ for $\Delta = 0$ Hz and the driving force amplitudes $F_{d1} = 0.21$ pN (blue) and $F_{d1} = 0.48$ pN (dark blue). The blue line is a fit to the linear regime. The dark blue line is obtained by multiplying the blue line by the drive amplitudes ratio 2.28. Inset: 3D plot of the measured ratio $a_1/F_{d1}$. **b** The characteristic width $\Gamma_{peak}$ of the spectral peak versus drive amplitude for linear friction (black, no sideband pumping), and for positive/negative nonlinear friction (red/blue, pumping at the red-/blue-detuned secondary sideband with $\Delta = 0$ Hz). The solid lines are the theory (Supplementary Note 4). Error bars represent ±1 s.e. **c** Bistability of forced vibrations due to negative nonlinear friction, $\Delta = -35$ Hz. The driving force amplitude is 0.70 pN. At the bifurcation points $\omega_L$ and $\omega_H$, the amplitude jumps down from the upper branch. Inset: with $\Delta = -1000$ Hz, nonlinear friction is negligible. Mode 1 displays a standard Duffing hysteresis for $F_{d1} = 3.5$ pN (black); at $F_{d1} = 0.70$ pN no hysteresis occurs (gray). **d** Measured (dots) and calculated stable (solid lines) and unstable (dashed lines) vibration amplitudes for $F_{d1} = 0.595$ pN (green) and 0.980 pN (yellow). Inset: zoom in on the isolated branch

driving force back to value in Fig. 4c, the system settles onto the upper branch instead of the lower one.

Figure 4d plots the measured stable (dots) and calculated stable and unstable vibrational states (solid and dashed lines, respectively) for different values of the driving force $F_{d1}$. Our calculations show that within the isolated loop, the state at higher amplitude is stable, and this is the state that is measured. The intermediate-amplitude state is unstable. As the drive amplitude is increased, the loop increases in size. Eventually, the lower end of the loop touches the top of the lower branch of the frequency response and the two merge into a single curve that resembles the shape of the response of a resonantly driven Duffing oscillator with linear friction. The corresponding critical bifurcation is qualitatively different from the familiar critical bifurcation for the Duffing oscillator. As we show in Supplementary Note 4, there are two merging saddle-node bifurcations with four rather than three branches of stationary vibrations merging together. More data and a detailed discussion of this unusual response are given in Supplementary Note 4. Furthermore, the different maximum vibration amplitudes of the isolated branch and the low-amplitude branch give rise to bistability in $\Gamma_{peak}$ as $F_{d1}$ is varied (Supplementary Note 5). The form of the hysteresis loop of the vibration amplitude as a function of the driving force differs from that of a Duffing oscillator (Supplementary Note 6).

## Discussion

Our findings show that by pumping two nonlinearly coupled microscale vibrational modes at the appropriate frequency, one can open a channel of decay involving two phonons of the slowly decaying mode and generate negative nonlinear friction in this mode. Such friction is controlled by the pumping and can be made sufficiently strong, so that the overall friction force becomes negative in a certain range of vibration amplitudes. This opens a new hitherto unexplored regime of the dynamics for micro/nanoscale resonators. As we show, the resulting instability is qualitatively different from the familiar case where the coefficient of linear friction is made negative, while nonlinear friction remains positive. For negative nonlinear friction, the zero-amplitude state always remains stable. Only when the system is perturbed by an extra pulse into a range of large amplitudes does it become unstable. It then settles into the regime of self-sustained vibrations with the amplitude at which the total friction force (averaged over the period) is zero.

Negative nonlinear friction qualitatively changes the dynamics of the mode even when it is not strong enough to lead to self-sustained vibrations. The effect is revealed in the response of the mode to a resonant driving force. Already for a relatively weak force, there emerges a large-amplitude branch of forced vibrations. As a function of the force frequency, this branch is disconnected from the coexisting Lorentzian-type small-amplitude branch. Accessing the large-amplitude branch requires activation to perturb the vibration amplitude to large values. Once the mode is on the large-amplitude branch, shifts in the mode frequency or the driving frequency can induce jumping back to the small-amplitude state. Unlike the driven Duffing oscillator, here jumping occurs for frequency changes in either directions.

The study of negative nonlinear friction is advantageous for distinguishing between the relaxation that comes from the coupling to a thermal reservoir with a large number of degrees of freedom and a broad excitation spectrum, on the one hand, and the relaxation that comes from the coupling to a vibrational mode with a comparatively fast decay rate, on the other hand. Our experiment demonstrates that it is the dynamical nature of the fast-decaying mode that ultimately prevents the system from a

runaway. Such runaway would have occurred if the negative nonlinear friction force came from pumping-induced coupling to a thermal reservoir and therefore kept increasing with the increasing vibration amplitude. For the coupled modes, the stabilization mechanism is the dependence of the mode eigenfrequencies on the vibration amplitudes. Because of this dependence, in combination with the finite spectral width of the fast-decaying mode, the coupled modes are eventually tuned away from resonance with the pump as their amplitudes increase. The pumping-induced negative nonlinear friction falls off, and for large vibration amplitudes the overall decay rate approaches the linear decay rate.

While the negative nonlinear friction in our experiment originates from the coupling to another mode in the same micromechanical resonator, the analysis applies to mechanical modes coupled to microwave or optical cavities as well. We emphasize that in contrast to the driving-induced linear friction[17], driving-induced nonlinear friction leads to a strongly non-Boltzmann distribution over the eigenstates of the mechanical modes. The quantum and classical nonequilibrium statistical physics associated with such distribution, including the possibility to significantly reduce or increase the relative population of higher-lying excited states, warrants further study. In a broader sense, the studied system is a classical analog of periodically driven quantum systems. Floquet dynamics of such systems has been attracting much attention in various contexts, from topological insulators to time crystals to thermalization far from equilibrium, see ref. [42] and references therein. Our system is interesting in that it displays features that have not been discussed so far in the context of the Floquet dynamics.

On the application side, the observed bidirectional jumping of the amplitude of resonant response due to negative nonlinear friction can be used to detect, with high sensitivity, small bipolar frequency perturbations. This suggests a qualitatively new type of a bifurcation amplifier. Achieving high sensitivity in such an amplifier relies on choosing a driving amplitude to yield a narrow frequency range for the isolated branch (see Supplementary Note 4 for how to prepare the system in the isolated branch). For the self-sustained vibrations that emerge for stronger negative nonlinear friction, an advantageous feature is the possibility to turn them on and off without changing system parameters, which leads to improved controllability. Experiments are under way to explore the phase diffusion of the self-sustained vibrations due to negative nonlinear friction. Preliminary analysis indicates that the phase diffusion of the two modes are strongly correlated in a manner which is, however, different from that of negative linear friction induced by pumping at the primary sideband. The frequency of the self-sustained vibrations can be made extremely stable using a feedback loop in which the phase of one of the modes is measured and the output is used to control the phase of the other mode, which extends the mechanism of stable frequency downconversion[30] to a different frequency range.

## Methods

**Transduction scheme.** Measurement was performed at a temperature of 4 K, pressure of $< 10^{-5}$ torr, and magnetic field of 5 T perpendicular to the substrate. Two electrodes are located underneath the plate. One way to excite mode 1 (the plate mode) is to apply an ac probe voltage (probe 1 in Fig. 1c) at frequency $\omega_{d1}$ close to $\omega_1$ on one of the electrodes to generate a periodic electrostatic force $F_{d1}\cos\omega_{d1}t$. This ac voltage is set to zero when studying self-sustained vibrations. Vibrations of the plate mode are measured by detecting the change in the capacitance between the top plate and the other electrode. A charge-sensitive amplifier is connected to the top plate, the output of which is connected to the input of a lockin amplifier. The vibration amplitudes of the in phase ($X(t)$) and out of phase ($Y(t)$) components with respect to the reference signal at $\omega_{d1}$ yields the complex vibration amplitude $v_1(t) = (m_1\omega_1/2C_{sc})^{1/2}[X(t) - iY(t)]$. For mode 2 (the beam mode), vibrations in the plane of the substrate can be electrostatically excited by applying an ac probe voltage (probe 2) at frequency $\omega_{d2}$ to the side gates. Motion of the beam is detected by recording the ac current

generated by the electromotive force as the beam vibrates in-plane in the presence of the perpendicular magnetic field.

**Data availability**. The data that support the findings of this study are available from the corresponding author on request.

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

## Acknowledgements

This work is supported by the Research Grants Council of Hong Kong SAR, China (Project No. 16301818). M.I.D. is supported by the National Science Foundation (DMR-1514591 and CMMI-1661618).

## Author contributions

H.B.C. and M.I.D. conceived the idea of the work. H.B.C. designed the experiments. M.I.D. developed the theory. X.D. performed the experiments and analyzed the data. X.D., H.B.C. and M.I.D. co-wrote the paper.

## Additional information

**Competing interests:** The authors declare no competing interests.

