## [Peer Review File · Nature Communications]

Reviewers' comments:

Reviewer #1 (Remarks to the Author):

The authors present experimental and theoretical investigation of behavior of two coupled mechanical resonators. They find a novel regime of negative nonlinear damping and demonstrate that it leads, in addition to a stable driven (almost) linear response of the oscillator, to an appearance of another stable branch which has a much higher amplitude. This occurs for red-detuned driving. The physical regime is clearly different from the Duffing oscillator. Whereas the effect has been demonstrated for mechanical oscillators, it is obviously relevant for oscillators of any nature provided suitable coupling can be engineered.

The authors present qualitative explanations of the phenomena, as well as quantitative theory. The theory is used to fit the experiment. The authors only show the steady state regimes of the oscillations (in Fig. 4). It would be more convincing to see transients regimes as well, to demonstrate how one stable solution evolves into another one, however, given excellent agreement with the theory I can recommend the publication in this form after the authors have clarified the point I outline below.

In Fig. 3, the measurement have been taken only up to the detuning of 20 Hz. What happens for higher detunings? Whereas this is not the regime of the main interest of the manuscript, and I do not ask for measurements to be performed for higher detunings, the authors have a theory which describes this regime. However, neither S2 nor S3 (relevant for this situation) show any plots. A theoretical plot (which can be in SI) referring to the full behavior of the system (does the middle branch approach zero at infinite positive detuning?) will be in order.

Reviewer #2 (Remarks to the Author):

The manuscript is devoted to a micro-mechanical realization of coupled driven nonlinear oscillators, which demonstrate negative non-linear friction. The latter should be distinguished from the linear negative friction - the well studied and extremely useful Van der Pol oscillator. The authors came up with a cute pumping setup, which allow to transform negative contribution to the non-linear part of the friction. They present a thorough and very clear theoretical analysis, supporting the idea and giving quantitative explanation to the observed behavior. To the best of my knowledge, this an interesting and novel idea, which was never explored before. Both experimental and theoretical part look rather convincing and nicely compliment each other.

I support publishing the manuscript in Nature Communications.

My only request has to do with the "Discussion" section. In its current form it mostly reiterates the abstract and main findings from the body of the paper. Only the very last paragraph provides some hints to possible applications of the observed phenomena and venues of further research. The manuscript can benefit from eliminating repetitions and using the space for a more detailed discussion, which is currently squeezed into the single last paragraph.

Response to Reviewer 1

We are grateful to Reviewer 1 for the helpful comments and for recommending publication of our paper after his/her comments will have been addressed.

1. Reviewer 1 suggested to show the results on the amplitude of the unstable branch of the self-sustained vibrations for detuning Δ much larger than shown in Fig. 3. We have added Fig. S2 which expands the range of Δ up to 500 Hz and the following description to Supplementary Note 3.

“Figure S2 shows the calculated amplitude of self-sustained vibrations as a function of pump detuning Δ . The range of Δ is expanded compared to the measurement and calculations shown in Fig. 3 of the main text. For the stable limit cycle (upper dark-colored curves), the amplitude increases with Δ . The unstable limit cycle (light-colored curves), on the other hand, shows a non-monotonic dependence of the amplitude on Δ . For large Δ , the amplitude of the unstable limit cycle increases with Δ . This behavior agrees with the notion that it requires larger perturbations to excite stationary self-sustained vibrations as the pump frequency deviates more from the red-detuned secondary sideband.”

2. Reviewer 1 mentioned that it would be more convincing to see the transient regimes and to demonstrate how one stable solution evolves into another one. We have added Fig. S5 to explain how the stable solution of the upper branch resembling a Duffing oscillator evolves into the isolated branch upon reducing the driving amplitude, and how a state on the isolated branch switches to the lower branch upon sweeping the driving frequency.

The corresponding explanation is added to Supplementary Note 4.

“Figure S5 demonstrates one way to access the disconnected branch of stable vibrations (isolated branch in green). We first set the driving amplitude to be sufficiently large so that the frequency response resembles that of a Duffing oscillator (orange). The driving frequency is increased toward the bistable region, ensuring that the system resides in the high-amplitude vibration state. Upon reaching the target frequency, the driving amplitude is then gradually lowered so that the system settles into the isolated branch (Fig. S5b). Once the system is on the isolated branch, if the driving frequency is increased (the green arrow that goes right in Fig. S5b) or decreased (the green arrow that goes left), at the corresponding bifurcation points the system switches to the low-amplitude branch.

We also added a discussion on the nature of the bifurcations in the main text.

“The corresponding critical bifurcation is qualitatively different from the familiar critical bifurcation for the Duffing oscillator. As we show in Supplemental Note 4, there are two merging saddle-node bifurcations with four rather than three branches of stationary vibrations merging together.”

Response to Reviewer 2.

We are grateful to Reviewer 2 for finding our paper to be interesting and novel, and for supporting the publication of our paper. Reviewer 2 suggested to provide a more detailed discussion of the results and to eliminate repetitions.

We have eliminated repetitive descriptions in the following places:

Abstract: “Isolated branches of self-sustained and forced vibrations are advantageous for applications in sensing and control.”

Introduction: “In contrast to self-sustained vibrations associated with negative linear friction [23-30], here the zero-amplitude state remains stable.”

Discussion: “In this latter case, it is well-known that the zero-amplitude state becomes unstable and the system starts oscillating.”

We have also expanded the last paragraph of possible applications and venues of further research (blue represents added text), and added reference 42:

“... We emphasize that, in contrast to the driving-induced linear friction [17], driving-induced nonlinear friction leads to a strongly non-Boltzmann distribution over the eigenstates of the mechanical modes. The quantum and classical nonequilibrium statistical physics associated with such distribution, including the possibility to significantly reduce or increase the relative population of higher-lying excited states, warrants further study. **In a broader sense, the studied system is a classical analog of periodically driven quantum systems. Floquet dynamics of such systems has been attracting much attention in various contexts, from topological insulators to time crystals to thermalization far from equilibrium, cf, [42] and references therein. Our system is interesting in that it displays features that have not been discussed so far in the context of the Floquet dynamics.**

On the application side, the observed bidirectional jumping of the amplitude of resonant response due to negative nonlinear friction can be used to detect, with high sensitivity, small bipolar frequency perturbations. This suggests a qualitatively new type of a bifurcation amplifier. **Achieving high sensitivity in such an amplifier relies on choosing a driving amplitude to yield a narrow frequency range for the isolated branch (see Supplementary Note 4 for how to prepare the system in the isolated branch).** For the self-sustained vibrations that emerge for stronger negative nonlinear friction, an advantageous feature is the possibility to turn them on and off without changing system parameters, which leads to improved controllability. **Experiments are under way to explore the phase diffusion of the self-sustained vibrations due to negative**

nonlinear friction. Preliminary analysis indicates that the phase diffusion of the two modes are strongly correlated in a manner which is, however, different from that of negative linear friction induced by pumping at the primary sideband. The frequency of the self-sustained vibrations can be made extremely stable using a feedback loop in which the phase of one of the modes is measured and the output is used to control the phase of the other mode, which extends the mechanism of stable frequency downconversion [30] to a different frequency range.”

REVIEWERS' COMMENTS:

Reviewer #1 (Remarks to the Author):

I am satisfied with the changes the authors introduced to the manuscript responding to the referees comments, and I can now recommend the manuscript for publication.

Reviewer #2 (Remarks to the Author):

I have read the authors response to both referees and found it fully sufficient. I thus support the publication of the manuscript in its current form.